# Structural mechanism of TRPV3 channel inhibition by the anesthetic dyclonine

Arthur Neuberger [1], Kirill D. Nadezhdin [1] & Alexander I. Sobolevsky [1✉]

Skin diseases are common human illnesses that occur in all cultures, at all ages, and affect between 30% and 70% of individuals globally. TRPV3 is a cation-permeable TRP channel predominantly expressed in skin keratinocytes, implicated in cutaneous sensation and associated with numerous skin diseases. TRPV3 is inhibited by the local anesthetic dyclonine, traditionally used for topical applications to relieve pain and itch. However, the structural basis of TRPV3 inhibition by dyclonine has remained elusive. Here we present a cryo-EM structure of a TRPV3-dyclonine complex that reveals binding of the inhibitor in the portals which connect the membrane environment surrounding the channel to the central cavity of the channel pore. We propose a mechanism of TRPV3 inhibition in which dyclonine molecules stick out into the channel pore, creating a barrier for ion conductance. The allosteric binding site of dyclonine can serve as a template for the design of new TRPV3-targeting drugs.

[1] Department of Biochemistry and Molecular Biophysics, Columbia University, New York, NY, USA. ✉email: as4005@cumc.columbia.edu

Transient receptor potential (TRP) channels constitute a super-family of cation-permeable ion channels that act as transducers of sensory modalities, including temperature, taste, olfaction, vision, hearing, and touch. TRP channel malfunctions are associated with numerous human diseases, including various forms of cancer. Vanilloid-subfamily member 3 TRP channel (TRPV3) is expressed in skin keratinocytes[1–4] as well as in corneal[5] and distal colon epithelial cells[6]. TRPV3 is implicated in cutaneous sensation, including temperature sensation, nociception, and itch[1–3,7–12]. The channel is also involved in the maintenance of a healthy skin barrier as well as in wound healing[5,13,14], hair growth[15–17], and embryonic development[18]. TRPV3 dysfunction has been associated with numerous skin diseases, including atopic dermatitis, dermal fibrosis, rosacea, and itch, as well as channelopathies like Olmsted syndrome, caused by a growing list of 'gain-of-function' mutations[19–26]. Overexpression of TRPV3 channels has also been implicated in the development and progression of various forms of cancer[27]. Correspondingly, inhibition of TRPV3 might be beneficial for the treatment of a multitude of diseases.

Natural and synthetic compounds, including osthole[26,28], isopentenyl pyrophosphate[29], farnesyl pyrophosphate[30], 17(R)-resolvin D1[31], and forsythoside B[32], have been identified as TRPV3 antagonists with varying prospects for clinical application. Among them, detailed structural information on the binding site and mechanism of TRPV3 inhibition has only been revealed for the natural, plant-derived antagonist osthole[28]. Recently, it has been reported that TRPV3 is also potently and selectively inhibited by dyclonine[33]. The piperidine dyclonine (1-(4-butoxyphenyl)-3-piperidin-1-ylpropan-1-one) is an aromatic ketone that is approved by the FDA as a clinical anesthetic and antipruritic agent for topical applications (0.5% or 1% dyclonine hydrochloride) to relieve pain and itch in patients suffering from ameliorating inflamed, excoriated, and broken lesions on mucous membranes and skin[34,35]. As an oral anesthetic, dyclonine is included in over the counter throat lozenges and sore throat spray products and is used to anesthetize mucous membranes for endoscopy[36]. Dyclonine has also been studied as a treatment of cancer[37,38] and Friedreich's ataxia, a rare inherited neurodegenerative movement disorder[39], as well as an anticonvulsant, multisynaptic spinal reflex depressant, and central nervous system stimulant[40]. Since other known TRPV3 inhibitors are not as potent and selective as dyclonine, this compound not only holds promise for medical applications but can also serve as a template for the design of new drugs. However, information about the molecular mechanism of dyclonine action on TRPV3, which is necessary for the structure-based drug design, has been missing.

To better understand the molecular mechanism of TRPV3 inhibition by dyclonine, we embarked on structural characterization of the TRPV3–dyclonine complex using cryo-electron microscopy (cryo-EM). We found that dyclonine binds to sites inside the portals which connect the membrane environment to the central cavity of the ion channel pore. These sites are formed by the transmembrane helices S5 and S6 and have not been previously shown to bind TRPV3 inhibitors. We confirm these sites using mutagenesis and measurements of calcium uptake, and propose a mechanism of inhibition in which dyclonine molecules stick out into the channel pore and create a barrier for ion conductance. Our structure provides a mechanistic understanding of TRPV3 inhibition by dyclonine and can serve as a template for the design of new drugs targeting TRPV3-linked diseases.

## Results and discussion

**Functional characterization and cryo-EM.** To study TRPV3 inhibition by dyclonine, we used mouse TRPV3, which shares 93% sequence identity with its human ortholog. We monitored TRPV3 inhibition by dyclonine using Fura-2-based measurements of changes in intracellular $Ca^{2+}$. Changes in the fluorescence intensity ratio at 340 and 380 nm ($F_{340}/F_{380}$) evoked by addition of 200 $\mu$M 2-APB were measured after pre-incubation of TRPV3-expressing HEK 293 GnTI$^-$ cells with various concentrations of dyclonine (Fig. 1a). Dyclonine inhibited TRPV3-mediated $Ca^{2+}$ uptake with the half-maximal inhibitory concentration, $IC_{50}$, of 29.8 ± 5.3 $\mu$M ($n = 7$, Fig. 1b).

We used cryo-EM to solve the structure of TRPV3 in complex with dyclonine, TRPV3$_{Dyc}$. Cryo-EM micrographs showed evenly dispersed TRPV3$_{Dyc}$ particles (Supplementary Fig. 1a), with diverse angular coverage (Supplementary Fig. 1b). High quality of the collected cryo-EM data was obvious from 2D class averages that showed clearly visible secondary structure elements (Supplementary Fig. 1c). Three-dimensional reconstruction with fourfold rotational symmetry (C4) resulted in a TRPV3$_{Dyc}$ map with overall resolution of 3.16 Å (Fig. 1c–f, Supplementary Figs. 1d, e, 2, Supplementary Table 1). For each subunit in the TRPV3$_{Dyc}$ homotetramer we built an accurate model of residues 118–756, excluding the regions of S1–S2 (residues 462–470) and S5-P (residues 611–621) loops that were not resolved clearly in the cryo-EM density.

**Structure and dyclonine binding site.** The overall architecture of TRPV3$_{Dyc}$ (Fig. 2) is similar to the previously solved TRPV3 structures[41–46]. The structure includes an ion channel, which is formed by the transmembrane domains (TMDs) of the four individual subunits, each contributing the S1–S4 and pore domains in a domain-swapped arrangement, and a large intracellular skirt that is mostly formed by the ankyrin-repeat domains (ARDs) (Fig. 2a, b). Each ARD is connected to the TMD by a linker domain. Amphipathic TRP helices run almost parallel to the membrane and interact with the linker domains and TMDs.

Upon inspection, the cryo-EM map of TRPV3$_{Dyc}$ revealed numerous non-protein densities but their majority represented annular lipids, typically observed around the TMD[41–46] (Fig. 1c, d). Exceptions were densities of the size of dyclonine molecules located in the side portals of the channel that connect the membrane environment to the pore's central cavity (Figs. 1e, f, 2a-c). Dyclonine can fit into these densities in two ways, with its butyl tail facing towards or away from the pore (Fig. 3). The two possible orientations can hardly be distinguished experimentally because of the missing density for the butyl tail. With a near-atomic local resolution of 2.7–2.8 Å (Supplementary Fig. 1d) and clear visibility of the adjacent protein side chains, the lack of the butyl tail density is likely due to its high flexibility. For illustration purposes, we chose the orientation with the outward-facing tails, which is consistent with the tertiary amine of dyclonine looking into the channel pore (Fig. 2d). This orientation is more probable for a significant fraction of dyclonine molecules that carry a positive charge (pKa 8.36) and are unlikely to insert their tertiary amine into the hydrophobic environment of the membrane.

The portal site in TRPV3 has not been previously shown to bind small-molecule inhibitors. This site, however, was originally proposed to be an entry pathway for local anesthetics to the pore of sodium channels[47–49] and found to accommodate cannabidiol in TRPV2, a closely-related representative of the vanilloid subfamily of TRP channels[50]. To verify binding of dyclonine to the portal site in TRPV3, we mutated the contributing residues Y594, F633, I663, and F666, and tested inhibition of the mutant channels by dyclonine using Fura-2-based measurements of changes in intracellular $Ca^{2+}$ in response to 2-APB application (Fig. 2e, f). For the Y594A mutant, the control application of 2-APB resulted in barely detectable changes in the fluorescence intensity ratio $F_{340}/F_{380}$, suggesting reduced efficacy of the agonist

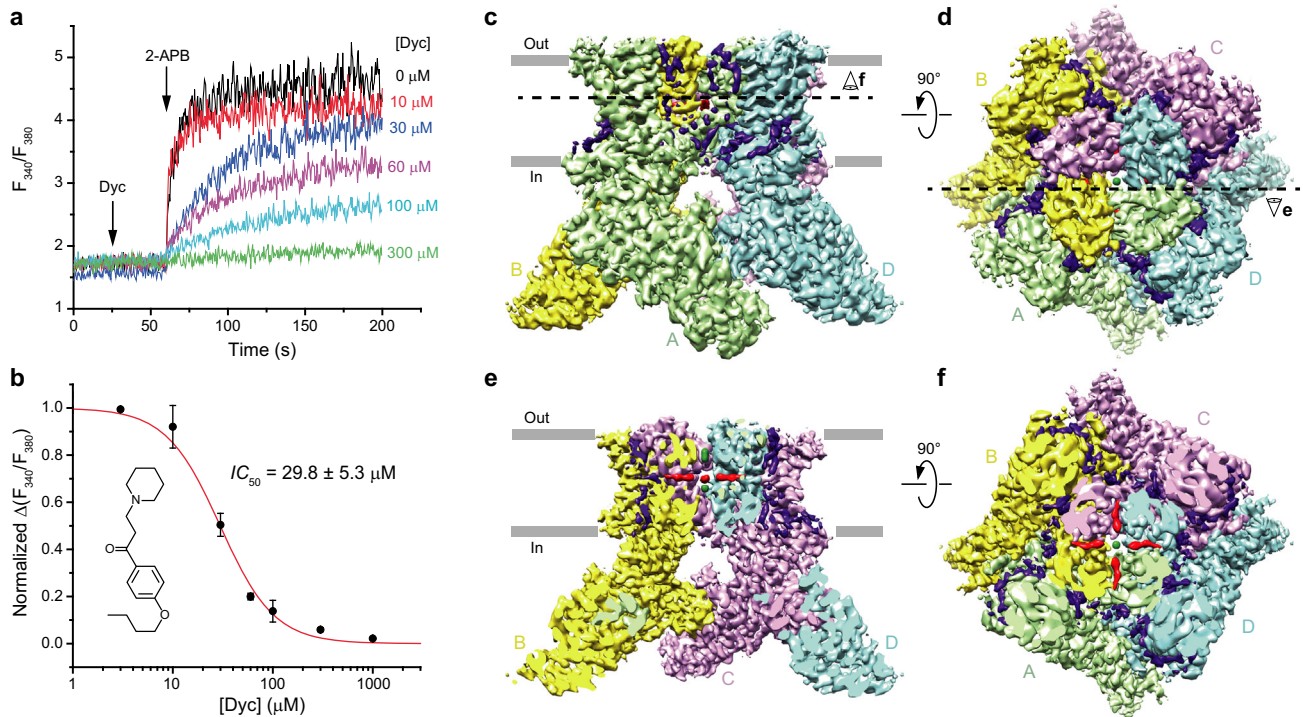

**Fig. 1 Dyclonine inhibition and cryo-EM. a** Representative ratiometric Fura-2-based fluorescence measurements of changes in intracellular $Ca^{2+}$ for HEK 293 $GnTI^{-}$ cells expressing wild-type mouse TRPV3. The changes in the fluorescence intensity ratio at 340 and 380 nm ($F_{340}/F_{380}$) were monitored in response to the addition of 200 μM 2-APB (arrow) after pre-incubation of cells with various concentrations of dyclonine. The experiment was repeated independently seven times with similar results. **b** Dose–response curve for TRPV3 inhibition by dyclonine. The changes in the $F_{340}/F_{380}$ ratio were normalized to its maximal value in the absence of dyclonine and fitted by the logistic equation (red line), with $IC_{50} = 29.8 \pm 5.3$ μM and $n_{Hill} = 1.61 \pm 0.27$ ($n = 7$ independent experiments). The values are mean ± SEM. The inset shows the chemical structure of dyclonine. Source data are provided as a Source Data file. **c, d** 3D cryo-EM reconstruction of $TRPV3_{Dyc}$ viewed from the side (**c**) or top (**d**), with subunits colored green, yellow, purple, and cyan. **e, f** The same cryo-EM density as in (**c, d**), but cut off along the dashed lines in (**d**) and (**c**), respectively. Putative densities for dyclonine and sodium ions are shown in red and green.

and making assessment of dyclonine inhibition impossible. We concluded that the Y594A mutation altered gating properties of TRPV3 and excluded this mutant from further analysis. For the other three mutants, control applications of 200 μM 2-APB resulted in $F_{340}/F_{380}$ values comparable to wild-type channels. In one of these, I663W, the mutation was introduced in an attempt to interfere with binding of dyclonine to the portal site. Unfortunately, the $IC_{50}$ value for I663W ($31.5 \pm 1.6$ μM, $n = 3$) was similar to wild-type channels, suggesting that this mutation did not significantly alter dyclonine binding, likely because the side chain of the introduced tryptophan adapts a conformation that does not interfere with dyclonine binding.

In contrast, the $IC_{50}$ value for dyclonine inhibition of the F666A mutant ($673 \pm 37$ μM, $n = 3$) was more than 20 times larger than the corresponding value for wild-type channels (Fig. 2f), consistent with the results of previous electrophysiological experiments[33] and strongly supporting reduced affinity of the F666A mutant to dyclonine and direct involvement of F666 in dyclonine inhibition. The effect of F666A was not due to disruption of the normal gating function because the concentration dependence of 2-APB-induced activation of F666A mutant channels ($EC_{50} = 20.7 \pm 0.9$ μM, $n = 3$) was similar to wild-type channels ($EC_{50} = 27.4 \pm 4.5$ μM, $n = 4$)[42] (Fig. 4a). Drug-independent changes introduced by the F666A mutation were also unlikely because inhibition of the F666A channels by the structurally distinct antagonist osthole that binds to sites at the base of S1-S4 and at the ARD-TMD linker[28], distal from the dyclonine portal site, had a potency ($IC_{50} = 33.1 \pm 3.8$ μM, $n = 3$) comparable to wild-type channels ($IC_{50} = 20.5 \pm 0.5$ μM, $n = 4$) (Fig. 4b).

Interestingly, the alanine substitution of F633, which guards the entry to the portal site from the membrane site, resulted in a three times smaller value of $IC_{50}$ for F633A ($9.8 \pm 0.7$ μM, $n = 6$) compared to wild-type channels (Fig. 2f). The increased affinity of the F633A mutant to dyclonine is therefore consistent with the small side chain of alanine creating an easier access for dyclonine to reach the portal site than the bulky side chain of phenylalanine. As a negative control, we tested dyclonine inhibition of channels with the Y564A mutation in the agonist binding site at the base of the S1-S4 helical bundle, which is distal from the putative dyclonine binding sites. The Y564A mutation increases affinity to the TRPV3 agonist 2-APB[42] and decreases affinity to the competitive antagonist osthole[28]. Nevertheless, $IC_{50}$ for dyclonine inhibition of the Y564A mutant ($31.5 \pm 1.0$ μM, $n = 4$) was the same as $IC_{50}$ for wild-type channels.

We also ruled out the competitive inhibition of TRPV3 by dyclonine by measuring the concentration dependence of TRPV3 activation by 2-APB at two different concentrations of dyclonine. The resulting 2-APB concentration dependencies in double logarithmic coordinates (Schild plots[51]) showed clearly different slopes, emphasized by an intersection of lines that fit these dependencies and strongly supporting the lack of obvious competition between 2-APB and dyclonine (Fig. 4c). These results are in agreement with the observation that dyclonine also inhibits TRPV3 currents evoked by heat[33]. Therefore, our mutagenesis experiments combined with ratiometric measurements of intracellular $Ca^{2+}$ strongly support binding of dyclonine in the TRPV3 portal sites.

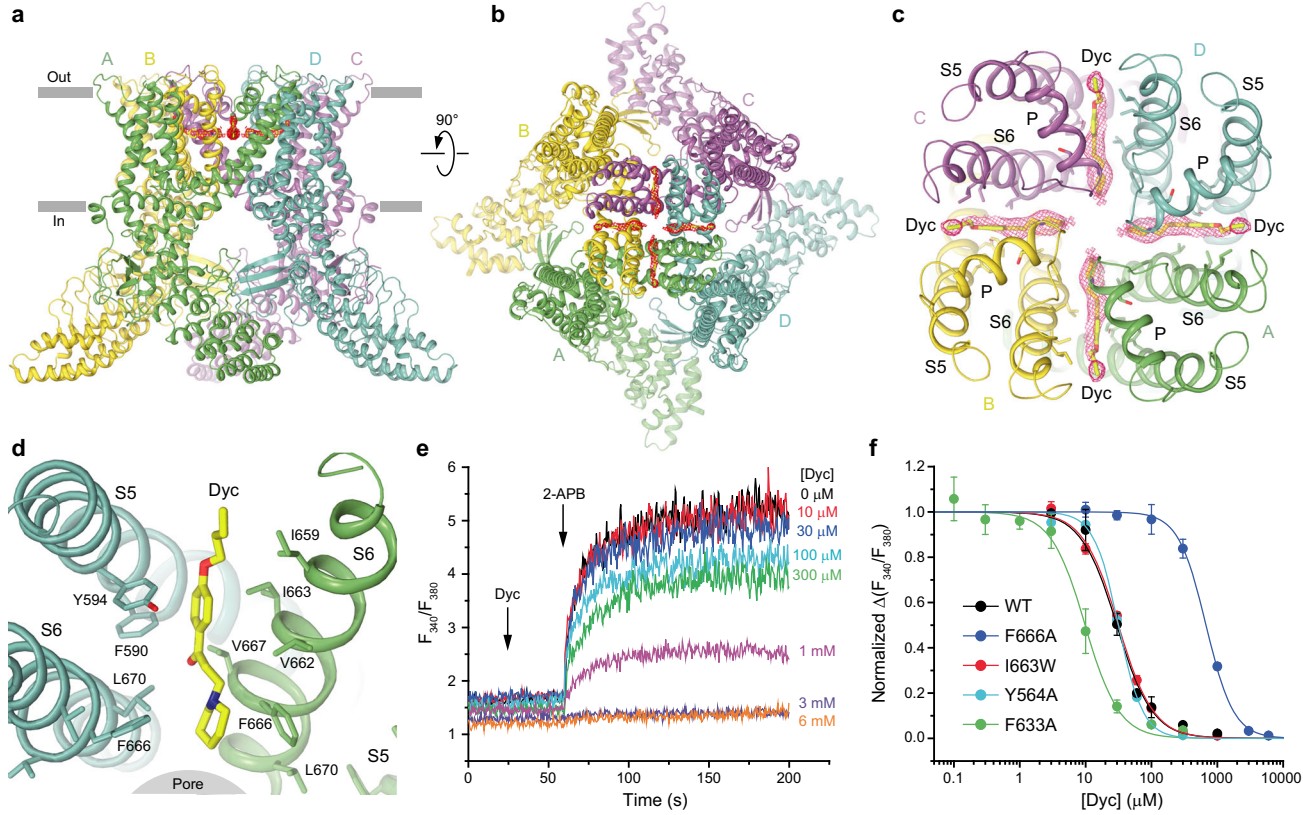

**Fig. 2 TRPV3_Dyc structure and dyclonine binding sites. a, b** TRPV3_Dyc structure viewed from the side (**a**) or top (**b**), with subunits colored green, yellow, purple, and cyan. Red mesh shows densities for dyclonine. Dyclonine molecules are shown as sticks. **c, d** Close-up views of all four dyclonine binding sites (**c**) and only one of them (**d**). **e** Representative ratiometric fluorescence measurements of changes in intracellular $Ca^{2+}$ for HEK 293 GnTI⁻ cells expressing F666A mutant TRPV3 channels. The changes in the fluorescence intensity ratio $F_{340}/F_{380}$ were monitored in response to the addition of 200 μM 2-APB (arrow) after pre-incubation of cells with various concentrations of dyclonine. The experiment was repeated independently three times with similar results. **f** Dose–response curves for inhibition of wild-type and mutant TRPV3 channels by dyclonine. The changes in the fluorescence intensity ratio $F_{340}/F_{380}$ evoked by addition of 200 μM 2-APB after pre-incubation with various concentrations of dyclonine were normalized to their maximal values in the absence of dyclonine. Curves through the points are logistic equation fits, with $IC_{50} = 29.8 \pm 5.3$ μM and $n_{Hill} = 1.61 \pm 0.27$ ($n = 7$ independent experiments) for wild-type TRPV3, $IC_{50} = 673 \pm 37$ μM and $n_{Hill} = 2.07 \pm 0.08$ ($n = 3$ independent experiments) for F666A, $IC_{50} = 31.5 \pm 1.6$ μM and $n_{Hill} = 1.63 \pm 0.07$ ($n = 3$ independent experiments) for I663W, $IC_{50} = 31.5 \pm 1.0$ μM and $n_{Hill} = 2.33 \pm 0.11$ ($n = 4$ independent experiments) for Y564A and $IC_{50} = 9.8 \pm 0.7$ μM and $n_{Hill} = 1.68 \pm 0.19$ ($n = 6$ independent experiments) for F633A. Source data are provided.

**Conformational changes that accompany dyclonine binding.**
Conformational changes that accompany dyclonine binding were analyzed by comparing the TRPV3_Dyc structure with the previously solved closed-state structure of TRPV3 channel[52] (Fig. 5). To make room for dyclonine in the portal, the side chain of F666 flips around by about 90 degrees and its benzyl group becomes nearly parallel to the piperidine group of dyclonine. A similar flipping of the side chain of homologous Y634 has been observed in the portal of TRPV2 upon cannabidiol binding[50]. While other residues in S5 and S6 that surround dyclonine in TRPV3 (Fig. 2d) maintain similar conformations as in the closed apo state, the flipped up side chain of F666 causes a shift of the P-loop away from the portal, toward the extracellular space (Figs. 5a, 6a). This conformational change does not propagate to the channel gate, which is formed by the S6 helices bundle crossing and maintains a similar conformation as in the closed state. Concomitantly, the observed small movements in the selectivity filter induced by binding of dyclonine correlate with previous data, which suggest that mutations in this region might change dyclonine inhibition, presumably by allosterically interfering with the crucial F666 side chain[33] (Figs. 5a, 6).

**Mechanism and implications of TRPV3 inhibition by dyclonine.** Conformational changes that are critical for opening of TRPV3 occur below the π-bulge in S6 (Fig. 6a). In contrast,

dyclonine binding in the portals causes flipping of the F666 side chain and conformational changes in the P-loop above the π-bulge (Figs. 5, 6). Furthermore, dyclonine might inhibit TRPV3 closed-to-open conformational transition allosterically, for example, by stabilization of the closed state through altered interaction of permeant ions with the selectivity filter. The negative allosteric modulation of TRPV3 by dyclonine would be consistent with its reduced inhibitory potency recorded previously for the I674A gate residue mutant[33]. However, we think that it is more likely that dyclonine inhibits TRPV3 by binding in the side portals and protruding into the TRPV3 pore's central cavity, thus creating an additional barrier for ion permeation. This barrier can be hydrophobic if dyclonine molecules are neutral or can act through electrostatic repulsion when the inhibitor is positively charged. Permeant ions are therefore getting stopped by dyclonine in the pore like a Christmas tree trunk by thumb screws in a Christmas tree stand. This mechanism of TRPV3 inhibition by dyclonine should not be different depending on whether the channel is activated by an agonist or heat. Indeed, previous experiments with TRPV3 activation by an infrared laser demonstrated the lack of temperature-dependence of dyclonine inhibition[33].

There are two major ways of how dyclonine can reach its binding site in the side portal of the channel: through the (1) pore

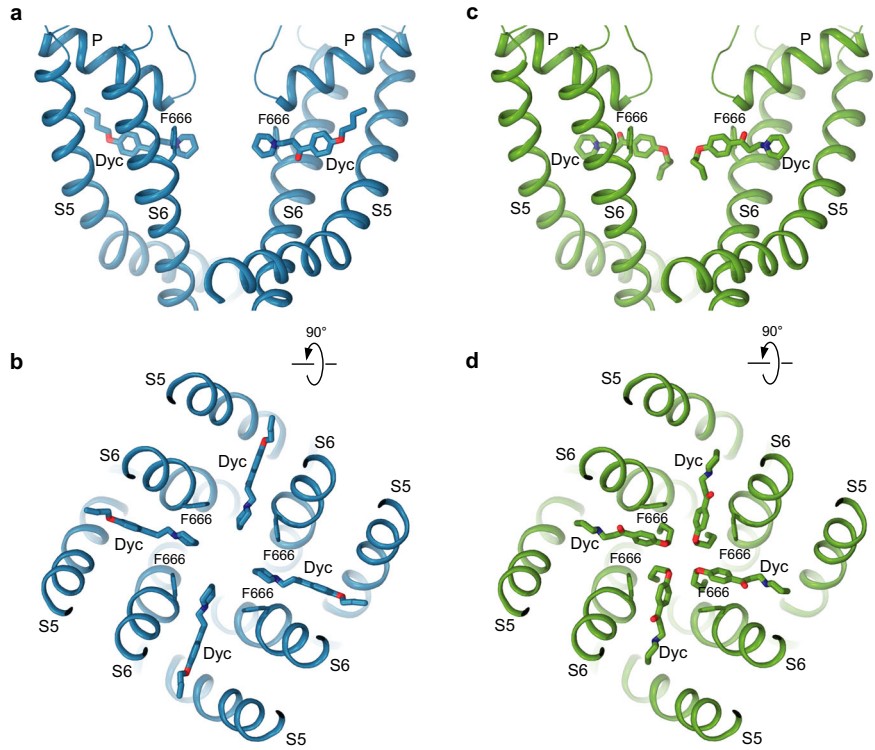

**Fig. 3 Two possible orientations of dyclonine in the portal site.** Pore domain in TRPV3$_{Dyc}$ with dyclonine in the head towards the pore (blue, **a**, **b**) or tail towards the pore (green, **c**, **d**) orientations, viewed parallel to the membrane (**a**, **c**) or extracellularly (**b**, **d**). The F666 side chains and dyclonine molecules are shown in sticks. Only two of four TRPV3 subunits are shown in (**a**) and (**c**), with the front and back subunits omitted for clarity.

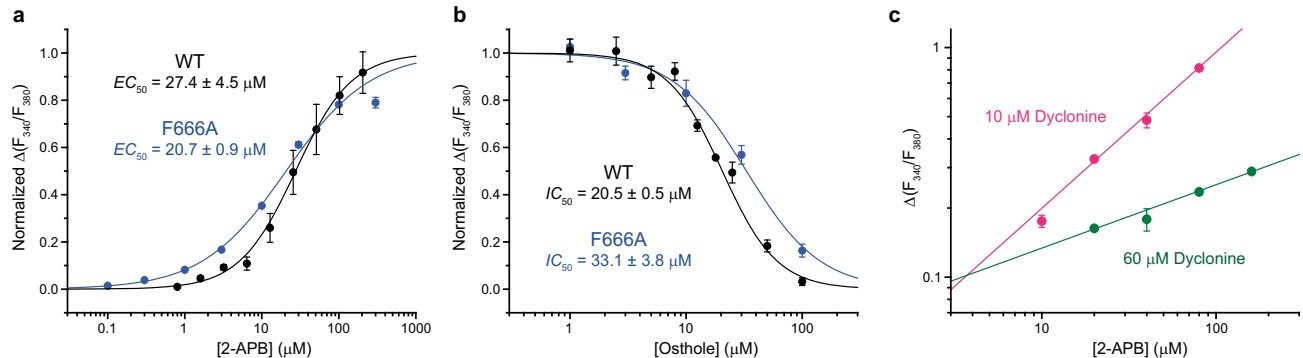

**Fig. 4 Control functional experiments. a** Dose–response curves for the F666A mutant and wild-type TRPV3 activation by 2-APB. The changes in $F_{340}/F_{380}$ were normalized to their approximated maximal values at saturating concentrations of 2-APB. Curves through the points are fits with the logistic equation and $EC_{50} = 27.4 \pm 4.5\,\mu M$ and $n_{Hill} = 1.19 \pm 0.09$ ($n = 4$ independent experiments) for wild-type TRPV3 and $EC_{50} = 20.7 \pm 0.9\,\mu M$ and $n_{Hill} = 0.80 \pm 0.02$ ($n = 3$ independent experiments) for F666A. The data for wild-type TRPV3 have been published before[1]. **b** Dose–response curves for inhibition of the F666A mutant and wild-type TRPV3 by osthole. The changes in $F_{340}/F_{380}$ were normalized to their maximal values in the absence of osthole. Curves through the points are fits with the logistic equation and $IC_{50} = 20.5 \pm 0.5\,\mu M$ and $n_{Hill} = 1.84 \pm 0.14$ ($n = 4$ independent experiments) for wild-type TRPV3 and $IC_{50} = 33.1 \pm 3.8\,\mu M$ and $n_{Hill} = 1.37 \pm 0.19$ ($n = 3$ independent experiments) for F666A. The data for wild-type TRPV3 have been published before[2]. **c** Double logarithmic Schild plot for 2-APB concentration dependencies of TRPV3 activation in the presence of 10 μM (pink circles, $n = 3$ independent experiments) and 60 μM (green circles, $n = 3$ independent experiments) of dyclonine. Lines through the points of the corresponding color are linear fits. For all panels, data points are presented as mean ± SEM and source data are provided.

and (2) membrane. Since pKa for dyclonine is 8.36, a significant fraction of the inhibitor molecules is positively charged, while the rest are neutral and carry no net charge. Interestingly, TRPV3 inhibition by dyclonine shows no voltage-dependence[33], indicating that the positively charged molecule is unlikely to reach the deep binding site through the selectivity filter of the channel pore. Instead, it might be easier for the neutral form of this drug to reach the portal site by entering the membrane[42]. Our hypothesis is that the neutral form of dyclonine approaches the portal sites

via the membrane pathway and acts in a way as originally proposed by Bertil Hille for local anesthetics that inhibit voltage-gated sodium channels[47]. In support of this hypothesis, mutations at the BM$_A$ site proposed by Liu et al., which is formed by the pore loop and S6 and located on the way of a dyclonine molecule traveling to the portal site, produce diverse effects on dyclonine inhibition[33]. Of course, some of these effects might be due to non-specific, transient or low-occupancy binding of dyclonine to the proposed BM$_A$ site, which was impossible to reveal using the

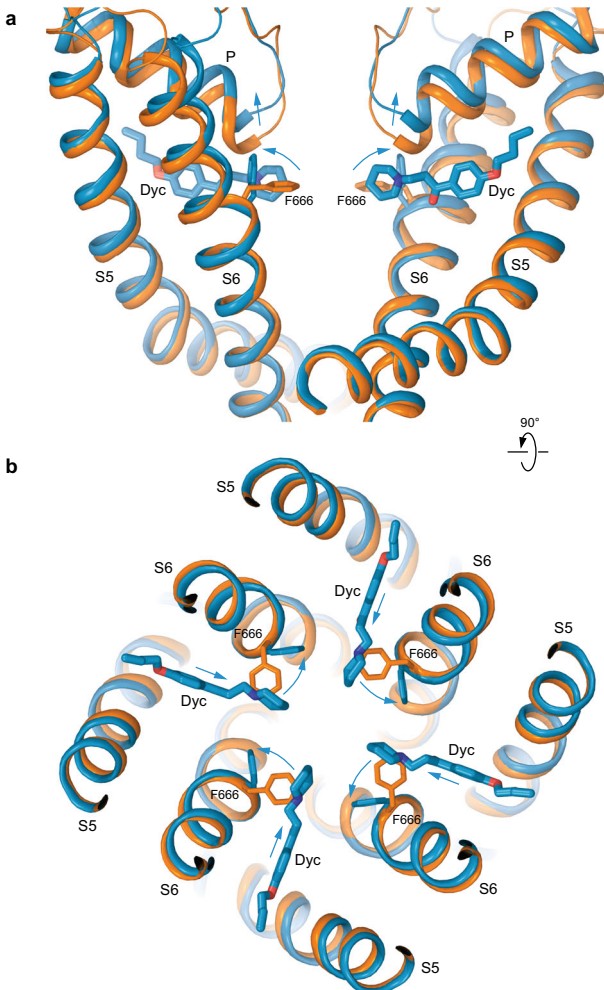

**Fig. 5 Conformational changes that accompany dyclonine binding.**
Superposition of the pore domains in TRPV3$_{Dyc}$ (blue) and apo-state TRPV3 (PDB ID: 7MIN; orange) viewed parallel to the membrane (**a**) or extracellularly (**b**). The F666 side chains and dyclonine molecules are shown in sticks. Only two of four TRPV3 subunits are shown in (**a**), with the front and back subunits omitted for clarity. Blue arrows indicate flipping of the F666 side chain, movement of P-loop extracellularly and of dyclonine towards the pore center upon binding in the transmembrane portals.

cryo-EM structure of TRPV3$_{Dyc}$. Obtaining structural evidence of such binding would require additional experimentation.

In summary, we present structural and functional data that reveal the binding site and suggest a mechanism of TRPV3 inhibition by the anesthetic dyclonine. This potent and selective inhibition of TRPV3 is also safe as dyclonine has been approved for clinical use. Correspondingly, the identified portal site can be further explored for rational drug design in search of new treatments for TRPV3-linked diseases.

## Methods

**Expression and purification.** Mouse TRPV3 was expressed and purified as previously described[28,41,42,53] with slight modification. Briefly, bacmids and baculoviruses were produced using a standard method[54]. Baculovirus was made in Sf9 cells (Thermo Fisher Scientific, mycoplasma test negative, GIBCO #12659017) for ~72 h and was added to suspension-adapted HEK 293 S cells lacking *N*-acetyl-glucosaminyltransferase I (GnTI⁻, mycoplasma test negative, ATCC #CRL-3022) that were maintained at 37 °C and 5% CO$_2$ in Freestyle 293 media (Gibco-Life Technologies #12338-018) supplemented with 2% FBS. To enhance protein expression, sodium butyrate (10 mM) was added 12 h after transduction and the temperature was reduced to 30 °C. At 48–72 h post-transduction, the cells were harvested by centrifugation at 5471 *g* for 15 min using a Sorvall Evolution RC centrifuge (Thermo

Fisher Scientific), washed in phosphate buffer saline (PBS) pH 8.0, and pelleted by centrifugation at 3202 *g* for 10 min using an Eppendorf 5810 centrifuge.

The cell pellet was resuspended in the ice-cold buffer containing 20 mM Tris pH 8.0, 150 mM NaCl, 0.8 μM aprotinin, 4.3 μM leupeptin, 2 μM pepstatin A, 1 μM phenylmethylsulfonyl fluoride (PMSF), and 1 mM β-mercaptoethanol (βME). The suspension was then supplemented with 2% (w/v) digitonin and cells were lysed at constant stirring for 2 h at 4 °C. Unbroken cells and cell debris were pelleted in an Eppendorf 5810 centrifuge at 3202 *g* and 4 °C for 10 min. Insoluble material was removed by ultracentrifugation for 1 h at 186,000 *g* in a Beckman Coulter centrifuge using a Type 45 Ti rotor. The supernatant was added to the strep resin, which was then rotated for 1 h at 4 °C. The resin was washed with 10 column volumes of wash buffer containing 20 mM Tris pH 8.0, 150 mM NaCl, 1 mM βME, and 0.01% (w/v) glyco-diosgenin (GDN), and the protein was eluted using the same buffer supplemented with 2.5 mM D-desthiobiotin. The eluted protein was concentrated to 0.5 ml using a 100-kDa NMWL centrifugal filter (MilliporeSigma™ Amicon™) and then centrifuged in a Sorvall MTX 150 Micro-Ultracentrifuge (Thermo Fisher Scientific) using a S100AT4 rotor for 30 min at 66,000 g and 4 °C before injecting it into a size-exclusion chromatography (SEC) column. The protein was purified using a Superose™ 6 10/300 GL SEC column attached to an AKTA FPLC (GE Healthcare) and equilibrated with the buffer containing 150 mM NaCl, 20 mM Tris pH 8.0, 1 mM βME, and 0.01% (w/v) GDN. The tetrameric peak fractions were pooled and concentrated using a 100-kDa NMWL centrifugal filter (MilliporeSigma™ Amicon™) to 3.7 mg/ml.

Circularized NW30 nanodiscs (cNW30) were prepared according to the standard protocol[55] and stored at −80 °C as ~2–3-mg/ml aliquots in the buffer containing 20 mM Tris pH 8.0 and 150 mM NaCl before usage. For nanodisc reconstitution, the purified protein was mixed with cNW30 nanodiscs and soybean lipids (Soy polar extract, Avanti Polar Lipids) at the molar ratio of 1:3:166 (TRPV3:cNW30:lipid). The lipids were dissolved in the buffer containing 150 mM NaCl and 20 mM Tris pH 8.0 to reach the concentration of 100 mg/ml, and subjected to 5–10 cycles of freezing in liquid nitrogen and thawing in a water bath sonicator. Dyclonine was added to the TRPV3:cNW30:lipid mixture at the concentration of 100 μM. The nanodisc mixture (500 μl) was rocked at room temperature for 1 h. Subsequently, 40 mg of Bio-beads SM2 (Bio-Rad), pre-wet in the buffer containing 20 mM Tris pH 8.0 and 150 mM NaCl, were added to the nanodisc mixture, which was then rotated for 1 h at 4 °C. After adding 40 mg more of Bio-beads SM2, the resulting mixture was rotated at 4 °C for another ~14 h. The Bio-beads SM2 were then removed by pipetting and nanodisc-reconstituted TRPV3 was purified from empty nanodiscs using SEC with a Superose 6 10/300 GL SEC column equilibrated with the buffer containing 150 mM NaCl, 20 mM Tris pH 8.0, 1 mM βME, and 100 μM dyclonine. Fractions of nanodisc-reconstituted TRPV3 were pooled and concentrated to 2.2 mg/ml using a 100-kDa NMWL centrifugal filter.

**Cryo-EM sample preparation and data collection.** CF 1.2/1.3, Au-50 (300-mesh) grids were used for plunge-freezing. Prior to sample application, grids were plasma treated in a PELCO easiGlow glow discharge cleaning system (0.39 mBar, 15 mA, "glow" for 25 s, and "hold" for 10 s). The nanodisc-reconstituted TRPV3 was supplemented with 100 μM more dyclonine 10 min prior to grid freezing. A Mark IV Vitrobot (Thermo Fisher Scientific) set to 100% humidity at 4 °C was used to plunge-freeze the grids in liquid ethane after applying 3 μl of protein sample to their gold-coated side using a blot time of 5 s, a blot force of 5, and a wait time of 15 s. The grids were stored in liquid nitrogen before imaging.

Images of frozen-hydrated particles of TRPV3$_{Dyc}$ were collected using EPU 2 on a Titan Krios transmission electron microscope (Thermo Fisher Scientific) operating at 300 kV and equipped with a post-column GIF Quantum energy filter and a Gatan K3 Summit direct electron detection camera (Gatan, Pleasanton, CA, USA). 6,856 micrographs were collected in the counting mode with an image pixel size of 0.855 Å across the defocus range of −0.8 to −2.0 μm. The total dose of ~60 e⁻ Å⁻² was attained by using the dose rate of ~13.75 e⁻ pixel⁻¹ s⁻¹ across 50 frames during the 2.5-s exposure time.

**Image processing and 3D reconstruction.** Data were processed in cryoSPARC[56]. Movie frames were aligned using the patch motion correction. Contrast transfer function (CTF) estimation was performed on non-dose-weighted micrographs using the patch CTF estimation. Subsequent data processing was done on dose-weighted micrographs. Following CTF estimation, micrographs were manually inspected and those with outliers in defocus values, ice thickness, and astigmatism as well as micrographs with lower predicted CTF-correlated resolution (higher than 5 Å) were excluded from further processing (individually assessed for each parameter relative to the overall distribution). The total number of 4,413,740 particles were picked using internally generated 2D templates and extracted with 4x binning (128-pixel box size). After several rounds of selection through 2D classification, 2,244,981 particles (representing 69 classes) were further 3D classified (heterogeneous refinement) into four classes. 1,030,127 particles representing the best class were re-extracted without binning (256-pixel box size) and further 3D classified. The final set of 87,876 particles representing the best class were subjected to homogenous, non-uniform, and CTF refinement. The reported resolution of 3.16 Å for the final map was estimated using the gold standard Fourier shell correlation (GSFSC). The local resolution was calculated with the resolution range estimated using the FSC = 0.143 criterion[57]. Cryo-EM density visualization was done in UCSF Chimera[58].

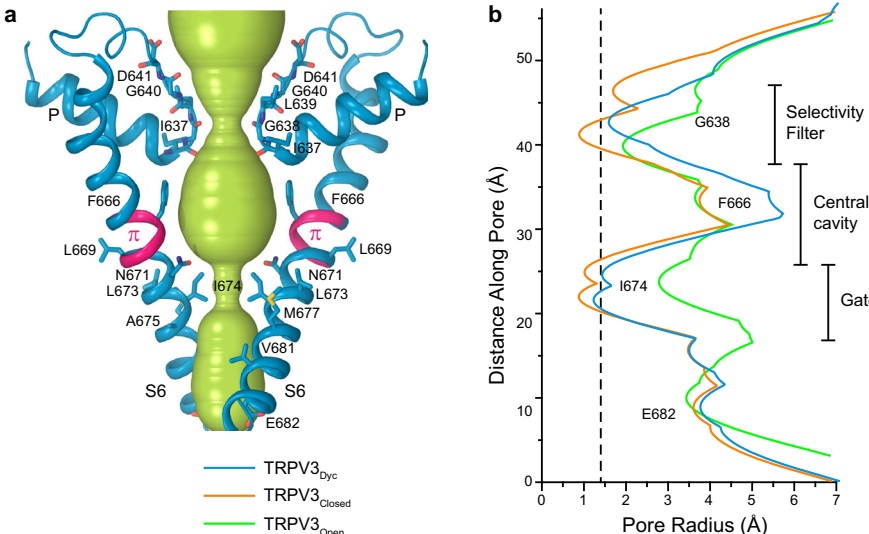

**Fig. 6 TRPV3$_{Dyc}$ pore and its radius in comparison with the closed and open states. a** Pore-forming domain in TRPV3$_{Dyc}$ with the residues contributing to pore lining in the dyclonine-bound (TRPV3$_{Dyc}$), closed (TRPV3$_{Closed}$) and open (TRPV3$_{Open}$) states shown as sticks. Only two of four subunits are shown, with the front and back subunits omitted for clarity. The pore profile is shown as a space-filling model (green). The region that undergoes α-to-π transition in S6 is highlighted in pink. **b** Pore radius for TRPV3$_{Dyc}$ (blue), TRPV3$_{Closed}$ (orange, PDB ID: 7MIN) and TRPV3$_{Open}$ (green, PDB ID: 7MIO) calculated using HOLE. The vertical dashed line denotes the radius of a water molecule, 1.4 Å.

**Model building**. To build the model of TRPV3 in Coot[59], we used the previously published cryo-EM structures of TRPV3[52] as guides. The model was tested for overfitting by shifting its coordinates by 0.5 Å (using Shake) in Phenix[60], refining the shaken model against the corresponding unfiltered half map, and generating a density from the resulting model in UCSF Chimera. The structure was visualized and figures were prepared in UCSF Chimera, UCSF ChimeraX[61], and Pymol[60]. The pore radius was calculated using HOLE[62].

**FURA-2 measurements**. GFP-free C-terminally strep-tagged wild-type TRPV3, TRPV3-Y564A, TRPV3-Y594A, TRPV3-F633A, TRPV3-I663W, and TRPV3-F666A were expressed in HEK 293 S GnTI⁻ cells as previously described[41,42,53]. In short, 48 h after transduction, 1-ml aliquots of cell suspension were collected by centrifugation at 550 g for 5 min using an Eppendorf 5424 centrifuge. The cells were resuspended in the pre-warmed modified HEPES-buffered saline (HBS) containing 140 mM NaCl, 4.8 mM KCl, 1 mM MgCl$_2$, 5 mM D-glucose, and 10 mM HEPES pH 7.4 and 5 μg/ml Fura-2 AM (Thermo Fisher Scientific) and incubated for 45 min at 37 °C. The Fura-2-loaded cells were then centrifuged for 5 min at 550 g, resuspended in the pre-warmed, modified HBS and incubated again for 30 min at 37 °C in the dark. The cells were subsequently pelleted and washed twice in the modified HBS. The washed cell pellets were kept on ice in the dark before taking an aliquot for fluorescence measurements. Immediately prior to each measurement, the cell pellets were resuspended in 2 ml of modified HBS and supplemented with 50 μM CaCl$_2$ and dyclonine at various concentrations. We used the lower CaCl$_2$ concentration than before[42,28] (2.5 mM) to minimize non-specific effects, which often appeared as a drift in the Fura-2 signal at high Ca$^{2+}$ concentration, presumably due to cell death as a result of calcium overload. Fluorescence measurements were conducted using a QuantaMaster 40 spectrofluorometer (Photon Technology International) at room temperature in a quartz cuvette under constant stirring. After 60 s since the beginning, the recording was paused and 2-APB was added. After the addition of 2-APB, the recording was continued for the additional 140 s. Changes in intracellular Ca$^{2+}$ were measured by taking the ratio of fluorescence emitted at 510 nm at the excitation wavelength of 340 nm and 380 nm. The excitation wavelength was switched at 200 ms intervals. Δ(F340/F380) was measured as the difference between baseline F340/F380 ratio recorded before the addition of dyclonine and the maximum F340/F380 after it has been added. The data were processed using Origin 9.0 (OriginLab) software.

**Reporting summary**. Further information on research design is available in the Nature Research Reporting Summary linked to this article.

## Data availability
Data that support the findings of this study are available from the corresponding author upon reasonable request. The cryo-EM density map has been deposited to the Electron Microscopy Data Bank (EMDB) under the accession code EMD-26488 (dyclonine-bound TRPV3). The atomic coordinates have been deposited to the Protein Data Bank (PDB)

under the accession code 7UGG [https://doi.org/10.2210/pdb7UGG/pdb] (dyclonine-bound TRPV3). Source data are provided with this paper.

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

## Acknowledgements

This research was, in part, supported by the National Cancer Institute's National Cryo-EM Facility at the Frederick National Laboratory for Cancer Research under contract HSSN261200800001E. A.N. is a Walter Benjamin Fellow funded by the Deutsche Forschungsgemeinschaft (DFG, German Research Foundation)–464295817. A.I.S. was supported by the NIH (R01 AR078814, R01 CA206573, R01 NS083660, R01 NS107253) and NSF (1818086).

## Author contributions

A.N. made constructs, prepared protein samples, and carried out cryo-EM data processing. A.N. and K.D.N prepared cryo-EM samples and carried out Fura-2 intracellular calcium imaging. A.N. and A.I.S. analyzed structural data and built molecular models. A.N. and A.I.S wrote the manuscript, which was then edited by K.D.N.

## Competing interests

The authors declare no competing interests.
