## [Peer Review File · Nature Communications]

Structural mechanism of TRPV3 channel inhibition by the anesthetic dyclonineReviewers' Comments:

Reviewer #1:

Remarks to the Author:

TRPV3 is an intensively studied ion channel implicated in various physiological processes; its involvement in skin health is particularly noticeable, as highlighted by strong skin and hair phenotypes linked to TRPV3 genetic mutations. Several cryo-EM structures of TRPV3 have been reported so far. The Sobolevsky group now presents in a succinct manuscript a new cryo-EM structure of TRPV3 in complex with antagonist dyclonine. This compound and its derivative are FDA-proved clinical anesthetic and antipruritic agent, however, the structural basis for their activity is not explained. Therefore, this is an important study.

The unexpected finding is the location of bound dyclonine. In the cryo-EM structure, dyclonine molecules bind at para-pore positions, with direct interactions with S5 and S6. This is an unusual ligand binding position in TRPV3 and most related TRP channels, though it has resemblance to that of certain Nav regulating molecules. Minor structural changes identified in the new structure suggest that binding of dyclonine may not terminate ion permeation by inducing conformational changes. Instead, it is suggested that bound dyclonine molecules protrude into the ion permeation pathway to interrupt ion permeation. This is the most reasonable interpretation based on the structural observations. However, a close correlation between pore diameter and open/closed states has been exaggerated in recent years. I would suggest that cautions should be taken when ruling out a gating conformational change, especially given that the observed changes in the selectivity filter may alter interactions of permeant ions with the pore.

For functional tests of point mutants, 2-APB was used as a control to confirm normal gating function. Results from these supportive measurements should be presented.

Jie Zheng

Reviewer #2:

Remarks to the Author:

The report describes a new cryo-EM structure of TRPV3 in complex with a recently described antagonist, dyclonine. It was found that dyclonine was wedged in the portal site of TRPV3 below the pore helix between the S5 helix of one protomer and the S6 helix of the neighboring protomer. It was proposed that the blocker could stick out into the channel pore to form a hydrophobic barrier for ion conductance. This suggests a new mode of channel regulation by small molecules. However, more work is needed in order to truly establish this mechanism.

1) The resolution, 3.16 Å, is not high enough to resolve dyclonine in the structure. The only thing that could be seen, or referred to, was the density of the whole molecule. Not only the bonds could not be resolved, the orientation of the drug was also uncertain. This makes the mutagenesis data difficult to interpret. The exclusion of S5-P (residues 611-621) loop from the structure adds more uncertainty as it is quite close to the identified site.

2) Previously, the dyclonine-binding site was predicted to be near the identified site by *in silico* docking, although not exactly at the same orientation and positions (Liu et al., 2021 eLife). Mutations were also made at residues that are near the new binding site identified here, with functional outcomes not too different from those shown in the current study, i.e., both increases and decreases in IC50 values with different mutations. What would be the explanation for the previous data if the new binding site is right and the previously suggested site is wrong? Interestingly, F666A was also tested in the previous study and shown to have a dramatically increased IC50. This should be noted in the current paper, especially because the current paper relies solely on population based Ca²⁺ measurement for functional assessment, which in my view is also a major shortcoming.

- 3) For all mutations, the inhibition by a structurally distinct antagonist that acts a different binding site needs to be tested in order to rule out any drug independent structural changes that could jeopardize the inhibitory action of the blockers.
- 4) What was the rationale behind testing I663W and F633A? Does the structure suggest that they are somewhat involved in interacting with dyclonine? If they do, in which way(s)? Do the functional results support the prediction?
- 5) The authors keep referring Fura2-AM as the indicator for Ca²⁺ measurement. This is incorrect as the AM-ester of Fura2 does not indicate Ca²⁺ levels. Only the de-esterified Fura2 does.
- 6) What was the reason for the low Ca²⁺ concentration (50 μM) used in the Ca²⁺ measurement?
- 7) Lines 142-143, should a decrease in IC₅₀ to an inhibitor be viewed as the mutant having a "reduced affinity" or an increased affinity?
- 8) Line 165, "close state" should be "closed state".

Reviewer #3:

Remarks to the Author:

Neuberger et al. tried to find the structural mechanism of TRPV3 inhibition by dyclonine using a cryo-EM, and proposed a mechanism in which dyclonine sticks out into the channel pore, creating a hydrophobic barrier for ion permeation.

I don't find a new concept beyond the findings reported by Liu et al in eLife (2021) in which the authors found the key residues in the pore region of TRPV3 involved in the inhibition by dyclonine. The authors in this manuscript even did not do intensive discussion regarding the inhibition mechanisms proposed in the eLife paper.

I would like to raise several points to be addressed.

1. The authors did not discuss the inhibition of heat-evoked TRPV3 channel by dyclonine.
2. In the Ca-imaging experiments, it would be better to show the 2-APB-induced TRPV3 activities in the point mutants.
3. Even from the current cryo-EM study, it is not sure whether the portal site in TRPV3 is the only binding site for dyclonine.
4. It would be better to show the evidence to rule out the competitive inhibition of TRPV3 activity by dyclonine with 2-APB in the experiments with different concentrations of 2-APB.
5. It is not clear whether F666 is directly involved in the inhibition by dyclonine.
6. Line 164, it could not be concluded that TRPV3 gating is insulated from dyclonine since the previous study in eLife showed that I674A mutation at the gate region negatively affects the TRPV3 inhibition by dyclonine.

We are very thankful to all Reviewers for their excellent suggestions. To address the Reviewers' concerns and critiques, we have performed additional experiments and made changes in the manuscript with the details outlined in our responses below.

Reviewer #1 (Remarks to the Author):

TRPV3 is an intensively studied ion channel implicated in various physiological processes; its involvement in skin health is particularly noticeable, as highlighted by strong skin and hair phenotypes linked to TRPV3 genetic mutations. Several cryo-EM structures of TRPV3 have been reported so far. The Sobolevsky group now presents in a succinct manuscript a new cryo-EM structure of TRPV3 in complex with antagonist dyclonine. This compound and its derivative are FDA-proved clinical anesthetic and antipruritic agent, however, the structural basis for their activity is not explained. Therefore, this is an important study.

We are very thankful to Reviewer 1 for the appreciation and kind words about our work.

The unexpected finding is the location of bound dyclonine. In the cryo-EM structure, dyclonine molecules bind at para-pore positions, with direct interactions with S5 and S6. This is an unusual ligand binding position in TRPV3 and most related TRP channels, though it has resemblance to that of certain Nav regulating molecules. Minor structural changes identified in the new structure suggest that binding of dyclonine may not terminate ion permeation by inducing conformational changes. Instead, it is suggested that bound dyclonine molecules protrude into the ion permeation pathway to interrupt ion permeation. This is the most reasonable interpretation based on the structural observations. However, a close correlation between pore diameter and open/closed states has been exaggerated in recent years. I would suggest that cautions should be taken when ruling out a gating conformational change, especially given that the observed changes in the selectivity filter may alter interactions of permeant ions with the pore.

We absolutely agree with Reviewer 1. We have now mentioned in the text that dyclonine might inhibit TRPV3 closed-to-open conformational transition allosterically, for example, by stabilization of the closed state through altered interaction of permeant ions with the selectivity filter. The negative allosteric modulation of TRPV3 by dyclonine would be consistent with its reduced inhibitory potency recorded previously for the I674A gate residue mutant (Liu et al., eLife 2021). The corresponding changes have been introduced into the text (lines 189-193).

For functional tests of point mutants, 2-APB was used as a control to confirm normal gating function. Results from these supportive measurements should be presented.

To confirm normal gating of mutants, we measured the 2-APB activation curve for the F666A mutant, which showed the strongest effect on dyclonine inhibition, and compared it to the activation curve for wild-type TRPV3. Fitting of the concentration dependence of 2-APB-induced activation with the logistic equation ($EC_{50} = 27.4 \pm 4.5 \mu\text{M}$ and $n_{\text{Hill}} = 1.19 \pm 0.09$, $n = 4$ for wild-type TRPV3 and $EC_{50} = 20.7 \pm 0.9 \mu\text{M}$ and $n_{\text{Hill}} = 0.80 \pm 0.02$, $n = 3$ for F666A) clearly demonstrated that the F666A mutation has no strong effect on activation of TRPV3 by the agonist 2-APB. These new data have now been presented as panel **a** of the new Supplementary

Figure 4.

Reviewer #2 (Remarks to the Author):

The report describes a new cryo-EM structure of TRPV3 in complex with a recently described antagonist, dyclonine. It was found that dyclonine was wedged in the portal site of TRPV3 below the pore helix between the S5 helix of one protomer and the S6 helix of the neighboring protomer. It was proposed that the blocker could stick out into the channel pore to form a hydrophobic barrier for ion conductance. This suggests a new mode of channel regulation by small molecules. However, more work is needed in order to truly establish this mechanism. 1) The resolution, 3.16 Å, is not high enough to resolve dyclonine in the structure. The only thing that could be seen, or referred to, was the density of the whole molecule. Not only the bonds could not be resolved, the orientation of the drug was also uncertain. This makes the mutagenesis data difficult to interpret. The exclusion of S5-P (residues 611-621) loop from the structure adds more uncertainty as it is quite close to the identified site.

We thank Reviewer 2 for the constructive criticism. To further verify the inhibitory mechanism and binding site location for dyclonine, we carried out additional control functional experiments (new Supplementary Figure 4). We have also added a discussion of the previously proposed binding site of dyclonine and other possible mechanisms of dyclonine inhibition (lines 189-214).

2) Previously, the dyclonine-binding site was predicted to be near the identified site by in silico docking, although not exactly at the same orientation and positions (Liu et al., 2021 eLife). Mutations were also made at residues that are near the new binding site identified here, with functional outcomes not too different from those shown in the current study, i.e., both increases and decreases in IC₅₀ values with different mutations. What would be the explanation for the previous data if the new binding site is right and the previously suggested site is wrong? Interestingly, F666A was also tested in the previous study and shown to have a dramatically increased IC₅₀. This should be noted in the current paper, especially because the current paper relies solely on population based Ca²⁺ measurement for functional assessment, which in my view is also a major shortcoming.

Our hypothesis is that dyclonine approaches the portal sites through the membrane pathway and acts in a way originally proposed by Bertil Hille for local anesthetics that inhibit voltage-gated sodium channels (Hille, JGP 1977). If our hypothesis is correct, it explains why mutations at the BM_A site (Figure 8 in Liu et al., eLife 2021), which would be on the way of a dyclonine molecule traveling to the BM_C site, produce diverse effects on dyclonine inhibition. We also mention that some of these effects might be due to non-specific, transient or low-occupancy binding of dyclonine to the proposed BM_A site, which was impossible to reveal using the cryo-EM structure of TRPV3_{Dyc}. The corresponding discussion has been added to the text (lines 201-214). We are also thankful to Reviewer 2 for pointing out our obvious oversight. We have now mentioned the electrophysiological results of Liu et al. on F666A, which are highly consistent with our Fura-2-based results (line 140).

3) For all mutations, the inhibition by a structurally distinct antagonist that acts a different binding site needs to be tested in order to rule out any drug independent structural changes that could jeopardize the inhibitory action of the blockers.

Great suggestion. We have tested inhibition of the F666A mutant, which showed the strongest effect on dyclonine inhibition, by the structurally distinct antagonist osthole that binds to completely different sites on TRPV3, at the base of S1-S4 and at the ARD-TMD linker, distal from dyclonine (see Neuberger et al., EMBO Reports, 2021) and compared it with wild-type TRPV3 inhibition by osthole. Fitting of the concentration dependence of osthole-induced inhibition with the logistic equation ($IC_{50} = 20.5 \pm 0.5 \mu\text{M}$ and $n_{\text{Hill}} = 1.84 \pm 0.14$, $n = 4$ for wild-type TRPV3 and $IC_{50} = 33.1 \pm 3.8 \mu\text{M}$ and $n_{\text{Hill}} = 1.37 \pm 0.19$, $n = 3$ for F666A) clearly demonstrated that the F666A mutation has no strong effect on inhibition of TRPV3 by the structurally distinct antagonist osthole. These new data have now been presented as panel **b** of the new Supplementary Figure 4.

4) What was the rationale behind testing I663W and F633A? Does the structure suggest that they are somewhat involved in interacting with dyclonine? If they do, in which way(s)? Do the functional results support the prediction?

According to the TRPV3_{Dyc} structure, the residues I663 and F633 surround the putative dyclonine molecule. The rationale behind changing the bulky side chain of F633, which guards the entry to the side portal from the membrane site, to small alanine side chain was to make the access for dyclonine to reach the portal site easier. The increase in affinity of F633A to dyclonine compared to wild-type channels (Fig. 2f) strongly supports this view. The rationale behind testing I663W mutant was to try to interfere with binding of dyclonine to the portal site. Unfortunately, this mutation did not significantly alter dyclonine binding as the IC_{50} value for I663W appeared to be similar to wild-type channels (Fig. 2f). We still wanted to show this result because the mutagenesis approach does not always give the expected outcome. We think that instead of adapting the conformation that would prevent dyclonine binding that we wished for when designing this mutation, the side chain of the introduced W663 adapts a conformation that does not interfere with dyclonine binding. The corresponding explanations have been added to the text (lines 132-137, 150-154).

5) The authors keep referring Fura2-AM as the indicator for Ca²⁺ measurement. This is incorrect as the AM-ester of Fura2 does not indicate Ca²⁺ levels. Only the de-esterified Fura2 does.

We are thankful to Reviewer 2 for great suggestion. We corrected the text accordingly.

6) What was the reason for the low Ca²⁺ concentration (50 μM) used in the Ca²⁺ measurement?

We found that the high Ca²⁺ concentrations in the presence of TRPV3 agonists often produce non-specific effects, causing a drift in the baseline fluorescent signal at 340 and 380 nm, presumably due to cell death as result of calcium overload. We never observed such non-specific effects at 50 μM of Ca²⁺. Instead, Fura2 responses at 50 μM of Ca²⁺ always have high amplitude and the results of our experiments are highly consistent. The corresponding explanation has been added to the text (lines 324-327).

7) Lines 142-143, should a decrease in IC₅₀ to an inhibitor be viewed as the mutant having a “reduced affinity” or an increased affinity?

Thank you for catching up this obvious typo. The text has been fixed accordingly.

8) Line 165, “close state” should be “closed state”.

This typo has been fixed as well.

Reviewer #3 (Remarks to the Author):

Neuberger et al. tried to find the structural mechanism of TRPV3 inhibition by dyclonine using a cryo-EM, and proposed a mechanism in which dyclonine sticks out into the channel pore, creating a hydrophobic barrier for ion permeation.

I don't find a new concept beyond the findings reported by Liu et al in eLife (2021) in which the authors found the key residues in the pore region of TRPV3 involved in the inhibition by dyclonine. The authors in this manuscript even did not do intensive discussion regarding the inhibition mechanisms proposed in the eLife paper.

We have extended the discussion of our results in comparison with the results of Liu et al. (eLife 2021). Overall, the results of Liu et al. and ours are highly consistent. Moreover, using the *in silico* docking, Liu et al. proposed three dyclonine binding sites in TRPV3, BM_{A-C}, and one of these sites, BM_C, seems to exactly coincide with the site determined by our TRPV3_{Dyc} structure (see Figure 8B in Liu et al., eLife 2021). Interestingly, Liu et al. rejected their BM_C site by arguing that dyclonine is a positive charged alkaloid and to cause voltage-independent inhibition, it cannot reach the BM_C site through the channel pore. However, dyclonine is a neutral molecule that carries zero net charge and can easily enter the membrane. In fact, our hypothesis is that dyclonine approaches the portal sites through the membrane pathway (and therefore in voltage-independent manner) and acts in a way originally proposed by Bertil Hille for local anesthetics that inhibit voltage-gated sodium channels (Hille, JGP 1977). If our hypothesis is correct, it explains why mutations at the BM_A site (Figure 8 in Liu et al., eLife 2021), which would be on the way of a dyclonine molecule traveling to the BM_C site, produce diverse effects on dyclonine inhibition. The corresponding discussion has been added to the text (lines 189-214).

I would like to raise several points to be addressed.

1. The authors did not discuss the inhibition of heat-evoked TRPV3 channel by dyclonine.

Previously it has been shown that the heat-activated and ligand-activated open states as well as closed states at different temperatures are essentially identical (see, for example, references 41-44, 59). Therefore, independent of whether dyclonine affects the equilibrium between the closed and open states, its mechanism of inhibition of heat- or ligand-activated TRPV3 in the first approximation is the same. This conclusion is also supported by the lack of temperature-dependence of dyclonine inhibition measured using an infrared laser (Figure 4C in Liu et al., eLife 2021). The corresponding discussion has been added to the text (lines 197-200).

2. In the Ca-imaging experiments, it would be better to show the 2-APB-induced TRPV3 activities in the point mutants.

We thank Reviewer 3 for the suggestion. We have added examples of Fura2-based ratiometric fluorescence measurements in response to 2-APB application for the F666A mutant, which showed the strongest effect on dyclonine inhibition (new panel e in Figure 2). In addition, we measured the 2-APB activation curve for the F666A mutant and compared it to the activation curve for wild-type TRPV3. Fitting of the concentration dependence of 2-APB-induced activation with the logistic equation ($EC_{50} = 27.4 \pm 4.5 \mu\text{M}$ and $n_{\text{Hill}} = 1.19 \pm 0.09$, $n = 4$ for wild-type TRPV3 and $EC_{50} = 20.7 \pm 0.9 \mu\text{M}$ and $n_{\text{Hill}} = 0.80 \pm 0.02$, $n = 3$ for F666A) clearly demonstrated that the F666A mutation has no strong effect on activation of TRPV3 by the agonist 2-APB. These new data have now been presented as panel a of the new Supplementary Figure 4a.

3. Even from the current cryo-EM study, it is not sure whether the portal site in TRPV3 is the only binding site for dyclonine.

Reviewer 3 is absolutely right. Cryo-EM or crystal structures cannot provide a 100% proof of the existence of only one site. Other sites, non-specific, transient or low-occupancy, can surely exist but escape structure determination. For example, although our structure has not detected dyclonine bound the site proposed by Liu et al in eLife (2021), this binding site may nevertheless exist and bind a small fraction of dyclonine molecules. The corresponding statement has been added to the text (lines 210-214).

4. It would be better to show the evidence to rule out the competitive inhibition of TRPV3 activity by dyclonine with 2-APB in the experiments with different concentrations of 2-APB.

We thank Reviewer 3 for the excellent suggestion. We have measured concentration dependence of TRPV3 activation by 2-APB at two different concentrations of dyclonine and presented the corresponding data in the form of a double logarithmic Schild plot (new Supplementary Figure 4c). The resulting 2-APB concentration dependencies in double logarithmic coordinates showed clearly different slopes, emphasized by intersection of lines that fit these dependencies and strongly supporting the lack of obvious competition between 2-APB and dyclonine. Our result is

therefore in good agreement with Liu et al in eLife (2021), who concluded that dyclonine is not a competitive antagonist because it inhibited TRPV3 currents evoked by both 2-APB and heat. The corresponding information has been added to the text (lines 161-167).

5. It is not clear whether F666 is directly involved in the inhibition by dyclonine.

Based on flipping of the F666 side chain upon binding of dyclonine, supported by clear density in cryo-EM maps, and strong shift of the concentration dependence of dyclonine inhibition (Fig. 2f), we argue that F666 is directly involved in inhibition by dyclonine. The corresponding information has been added to the text (lines 138-149).

6. Line 164, it could not be concluded that TRPV3 gating is insulated from dyclonine since the previous study in eLife showed that I674A mutation at the gate region negatively affects the TRPV3 inhibition by dyclonine.

This is a very good point brought up by Reviewer 3. We have now mentioned the possible effect of dyclonine on TRPV3 gating and gave an example of I674A mutation when discussing the mechanism of dyclonine inhibition (lines 189-193).

Reviewers' Comments:

Reviewer #1:

Remarks to the Author:

all my previous requests have been satisfactorily addressed.

Reviewer #2:

Remarks to the Author:

The authors made changes and added new data in response to previous critiques. However, there is no improvement in the resolution of the ligand-bound structure, which in my opinion is critical for making certain the binding mechanism of the drug.

Also I appreciate the corrections made about Fura2-AM. However, I do not agree with the statement made about exciting Fura2 and Fura2-AM at 340 and 380 nm "at the excitation wavelength of 340 nm (excites Ca²⁺-bound Fura 2) and 380 nm (excites Ca²⁺-free Fura-2 AM)". This erroneous statement is unnecessary, but if you want to explain how Fura2 works, please make sure that you say it correctly.

Reviewer #3:

Remarks to the Author:

The authors generally well responded to my and other reviewers' concerns.

I have one question which should be addressed before being accepted.

The authors mentioned that dyclonine is a neutral molecule while Liu's paper believed it is a positive charged alkaloid. An aliphatic amine, in fact, carries a charge or remains neutral mostly depending on the physiological pH. It would be more convincing if the authors provide more information about properties (for example pKa) and pH environment of dyclonine. If my understanding is correct, dyclonine was dissolved in a buffer with pH 8 (might vary) in the method. If that is the case, most tertiary amines are likely uncharged but I believe some are protonated.

We are very thankful to Reviewers for their excellent suggestions. To address the Reviewers' comments, we have made changes in the manuscript with the details outlined in our responses below.

Reviewer #1 (Remarks to the Author):

all my previous requests have been satisfactorily addressed.

Reviewer #2 (Remarks to the Author):

The authors made changes and added new data in response to previous critiques. However, there is no improvement in the resolution of the ligand-bound structure, which in my opinion is critical for making certain the binding mechanism of the drug.

We think that in this case, visualization of the ligand is limited to greater extent by the dynamic nature of its binding rather than the overall resolution of the structure. In fact, with 3.16 Angstrom overall resolution, the local resolution around the binding pocket is 2.7-2.8 Angstrom. At this resolution, we clearly see the surrounding protein side chains. The corresponding mention of the local resolution has been added to the text (lines 117-119).

Also I appreciate the corrections made about Fura2-AM. However, I do not agree with the statement made about exciting Fura2 and Fura2-AM at 340 and 380 nm "at the excitation wavelength of 340 nm (excites Ca²⁺-bound Fura 2) and 380 nm (excites Ca²⁺-free Fura-2 AM)". This erroneous statement is unnecessary, but if you want to explain how Fura2 works, please make sure that you say it correctly.

We have corrected the sentence by deleting "(excites Ca²⁺-bound Fura 2)" and "(excites Ca²⁺-free Fura-2 AM)".

Reviewer #3 (Remarks to the Author):

The authors generally well responded to my and other reviewers' concerns. I have one question which should be addressed before being accepted.

The authors mentioned that dyclonine is a neutral molecule while Liu's paper believed it is a positive charged alkaloid. An aliphatic amine, in fact, carries a charge or remains neutral mostly depending on the physiological pH. It would be more convincing if the authors provide more information about properties (for example pKa) and pH environment of dyclonine. If my understanding is correct, dyclonine was dissolved in a buffer with pH 8 (might vary) in the method. If that is the case, most tertiary amines are likely uncharged but I believe some are protonated.

Reviewer #3 is absolutely right and this was an oversight on our part. The pKa of dyclonine is 8.36. Correspondingly at pH 8.0 used in our experiments, a significant fraction of dyclonine is positively charged, while the rest of molecules is neutral. We have therefore supplemented our

text with discussions outlining possible consequences of the dyclonine electric charge. One of the possible consequences is that for the positively charged dyclonine, the tail-out mode of binding, with the tertiary amine looking into the pore and butyl tail facing the hydrophobic environment of membrane, is more favorable and supports our preferred orientation. Second, we included the mention of the charged form of dyclonine when discussing possible routes, which the drug can take to reach its binding site. For example, while the neutral form can reach its binding site by penetrating the membrane and entering the side portals from the membrane side, the positively charged form is likely to reach its binding sites by traveling through the pore. The corresponding discussions have been added to the text (lines 122-124, 201-202, 209-216).